# Methods of In Situ Quantitative Root Biology

**DOI:** 10.3390/plants10112399

**Published:** 2021-11-06

**Authors:** Taras Pasternak, José Manuel Pérez-Pérez

**Affiliations:** 1Centre for BioSystems Analysis, BIOSS Centre for Biological Signalling Studies University of Freiburg, Institute of Biology II/Molecular Plant Physiology, 79104 Freiburg, Germany; 2Instituto de Bioingeniería, Universidad Miguel Hernández, 03202 Elche, Spain

**Keywords:** root apical meristem (RAM), lateral root (LR), cell cycle, nuclear structure, chromatin organization

## Abstract

When dealing with plant roots, a multiscale description of the functional root structure is needed. Since the beginning of 21st century, new devices such as laser confocal microscopes have been accessible for coarse root structure measurements, including three-dimensional (3D) reconstruction. Most researchers are familiar with using simple 2D geometry visualization that does not allow quantitative determination of key morphological features from an organ-like perspective. We provide here a detailed description of the quantitative methods available for 3D analysis of root features at single-cell resolution, including root asymmetry, lateral root analysis, cell size and nuclear organization, cell-cycle kinetics, and chromatin structure analysis. Quantitative maps of the root apical meristem (RAM) are shown for different species, including *Arabidopsis thaliana* (L.), Heynh, *Nicotiana tabacum* L., *Medicago sativa* L., and *Setaria italica* (L.) P. Beauv. The 3D analysis of the RAM in these species showed divergence in chromatin organization and cell volume distribution that might be used to study root zonation for each root tissue. Detailed protocols and possible pitfalls in the usage of the marker lines are discussed. Therefore, researchers who need to improve their quantitative root biology portfolio can use them as a reference.

## 1. Introduction

The root is an essential organ required for anchoring the plant within the soil/substrate, as well as for water and nutrient uptake. It is a rapidly growing organ (up to 15 mm/per day in *Arabidopsis thaliana* (L.), Heynh.) characterized by a small rootward (i.e., distal) region (including the columella and root cap (RC)) which protects the growing tip from mechanical damage during soil penetration, and a large shootward (i.e., proximal) region responsible for overall root growth [1]. The rootward region of the root also acts as a signaling hub for growth coordination in response to the soil environment [2].

The root apical meristem (RAM) is composed of dividing cells that are on a gradual transition from a stem-cell fate into a differentiated one [3]. Detailed investigation of the root structure was initiated more than 120 years ago, shortly after the building of the first optical microscopes. A detailed radial root structure with all cell origins was thoroughly described at that time in Ranunculaceae [4]. Later on, with the development of DNA labeling techniques, the temporal and spatial location of mitotic cells was investigated in *Allium cepa* L. root meristems [5,6]. Following these earlier studies, cell-cycle kinetics in the RAM was studied in pea (*Pisum sativum* L.) by quantifying the ratio of cells in a particular stage of the cell cycle to the total number of proliferating cells [7]. With the development of the thymidine incorporation assay [8], more data about cell cycle kinetics in the root system became available. Specifically, Clowes [9] and Van’t Hof and Sparrow [10] proposed a method based on ^3^H-thymidine incorporation into the replicating DNA. Van’t Hof further modified this method to accumulate cells in mitosis by additional colchicine treatment [11]. Recently, 5-ethynyl-2’-deoxyuridine (EdU) incorporation has been used to study cell-cycle kinetics in the root [12,13]. A detailed study of the embryonically derived root (so-called primary root (PR)) of the *A. thaliana* model was published by Dolan et al. [14] one century after the pioneering work of Maxwell [4].

Meristem zonation is another open question in root biology. It was earlier stated that the RAM consists of cells with different cell-cycle kinetics and different cell fates [9]. However, RAM length in *A. thaliana* is usually defined on the basis of the number of non-elongating cortex precursor cells from 2D images, without considering any association with their proliferation activity and without considering the different tissue fates [15], except a rarely used new algorithm based on 3D root analysis [16]. Recently, Salvi et al. [17] also defined the transition zone within the RAM according to cortex precursor cell length increase. However, it is not clear which cortex precursor cells should be used to define RAM size in other species such as wheat, tomato, and alfalfa, which contain up to five cortex layers. Here, we provide the rationale for the precise connection between cell volume and cell proliferation activity in the whole RAM.

The kinetics of the cell cycle and its organ-wide distribution do not receive enough attention in existing methods. Earlier work [18] demonstrated that cell-cycle duration is different in different cell types and dependent on cell fate. Thus, the widely used CyclinB1;1 marker does not adequately reflect all cells within the RAM that are in the G2/M transition phase because their expression is dependent, among other factors, on the G2 duration of each cell, which for example is longer in the epidermis precursor and the cortex precursor cells [19]. The marker-free method described here allows detecting cell-cycle and endocycle kinetics in every cell within the RAM.

The plant nucleus is not an inactive container of the DNA, and its dynamic organization plays a crucial role in coordinating DNA replication, DNA transcription, and cell fate. The plant nucleus shows specific cell fate-dependent and position-dependent structural, compositional, and functional features [20]. For example, specific characteristics of the nuclear shape and size, distribution and composition of nuclear domains, heterochromatin content, and chromatin condensation might serve as precise fingerprint of the cell fate and cell-cycle status.

The next gap in root biology is the characterization of the so-called rootward region of the root tip, including the RC and columella cells [1,2]. In addition to its protective and signaling role, the root tip may serve as an ideal model to study all cell development stages from cell division to terminal chromatin condensation, including cell death in the outer RC layers. Another missing feature in root biology studies is the detailed investigation of chromatin structure/organization in situ. Nuclear morphology is intrinsically linked to biological processes such as gene expression regulation, cell fate establishment, and cell differentiation [21]. Moreover, chromatin structure can also be used as a cell-cycle marker [22]. However, a robust and straightforward method for analyzing a large number of nuclei within the RAM and its link with cell fate and cell-cycle kinetics is missing. As already shown by Maxwell [4] and Dolan et al. [14], quiescent center (QC) cells have a very compact nucleus with a long (>24 h) G1 phase. Zhang et al. [23] claimed that cytokinin could induce QC cell division, but no additional evidence has been presented. To clarify this point, a detailed investigation of the cell-cycle kinetics in the QC region and a quantification of the nuclear structure with the spatial resolution are required.

The workflow proposed in this review allows such analyses to connect chromatin organization with cell-cycle stages and cell differentiation status at a whole-organ level. The workflow discussed here for in situ quantitative root biology can be applied to non-model plant species, including dicotyledonous and monocotyledonous plants, which largely differ in their root system architectures [24]. In this review, we provide a comprehensive description of cell biology approaches to fulfill most of the existing gaps in RAM biology and to characterize the root of non-model plants at a single-cell and subcellular resolution.

## 2. Results

Despite their simple structure, roots are complex organs, and we can quantify their morphology at different biological scales, ranging from proteins to organelles, single cells, and tissues. Hence, we divided the proposed workflow of analysis into four modules: (I) nuclei and nuclear structure, (II) cell-cycle analysis, (III) protein and protein complex localization, and (IV) cell and organ geometry analysis (Figure 1). We used the *XYZ* and root coordinate system based on distance from the QC, distance from the root axis, and angles [25], which allowed extracting positional information for each cell within a single root. This approach was applied to study nuclear, cellular, cell-cycle, and protein localization features in the RAM, the differentiation zone of the root, and the young lateral roots (LRs) of a few plant species. Moreover, all of these four modules can be combined; thus, the simultaneous extraction of cell geometry, cell cycle, and nuclear structure is feasible.

### 2.1. Module I: Chromatin Structure Quantification

The aim of module I is to provide a detailed chromatin functional atlas of the root. The studied features in this module allow dissecting all nuclear organization, including the origin of DNA replication, histone modification, and the ratio between hetero- and euchromatin.

First, we investigated nuclear organization in *A. thaliana* PRs (Figure 2a). We observed substantial differences in nuclear morphology between cell types, especially between RC, cortex, and epidermis precursors, and the more inner tissue layers (Figure 2b). To quantify these differences, we performed volumetric nuclear analysis. We found rapid divergence of the nuclear volume and nuclear heterogeneity between cell layers even in the division zone of the RAM, with higher nuclear volume in the cortex and epidermis precursors (Figure 2c). Next, we extended our analysis to individual nuclear features using the NucleusJ 2.0 software (see Section 4) that allowed us to extract 14 morphological parameters per nucleus: volume, surface area, sphericity, elongation, and chromocenter numbers and distribution, among others [26]. When studying cells of the same layer in the PR, we can distinguish different cell-cycle stages in the division zone of the RAM on the basis of nuclear volume and other morphological features such as chromocenter abundance (Figure 2d,e).

In addition to the RAM, the elongation and differentiation zones of the root play a key role for unveiling the mechanism of LR formation and nodule/pseudo-nodule induction. In the RAM, each cell has its own nuclear structure fingerprint, including nucleolus size, which varies according to its cell type and its position. By applying module I analysis, some of these features could be measured in an early LR primordium arising from the PR (Figure 2f and Appendix A), and a comparison with the nuclear features from the same layer in the PRs is also possible.

Staining with nuclear-specific dye for determining cell-cycle stages and nuclear features in roots is a marker-free strategy that could be used in non-model plants. We demonstrated that such analysis is feasible for alfalfa (*Medicago sativa* L.) RAMs, and we clearly distinguished different cell-cycle stages (G1, G2, and mitotic cells) on the basis of nuclear volume and other morphological features (Figure 3a–e). Next, we applied module I analysis to the mature zone of tobacco (*Nicotiana tabacum* L.) roots. In contrast to those found in the mature zone of *A. thaliana* PRs [19], cortex cells in this region of the tobacco PRs can divide and even induce pseudo-nodules [27].

Module I, in combination with module IV (see module IV below) allows us to link nuclear features with cell features (Figure 3), as well as to study cell volume, nuclear volume, and nuclear position and structure with respect to cell fate.

### 2.2. Module II: Cell-Cycle Analysis

Contributing new cells from the RAM is critical for root growth. Cell-cycle progression is highly and dynamically regulated at the chromatin level and dependent on cell fate [28]. In module II, we used a marker-free method based on the combination of two DNA staining methods to identify cell-cycle stages (Figure 4a–c): (a) EdU staining that marks DNA replication events, and (b) nuclear labeling with DAPI that allows scoring mitotic figures. Thus, we can distinguish cells with four different characteristics: (1) cells with very compact nuclei that prevent nucleosome/chromatin disassembly (i.e., with slower frequency of DNA replication) in QC, (2, 3) cells in G1 and G2 in the division zone of the RAM with more relaxed chromatin and with different nuclear volume (Figure 4d), and (4) cells with irregular chromatin in differentiation and mature zones of the RAM. Incubation with EdU at variable times allows an investigation of the kinetics of cell-cycle progression, as a short EdU incubation (20–30 min) marks cells which currently replicate DNA (Figure 4e), while longer EdU incubation allows detecting S + G2 duration by quantification of an EdU-positive mitotic plate, and very long incubation (16–24 h) allows us to detect EdU-negative cells with long G1 (Appendix A), such as those of the QC that have low DNA replication rates [13].

Module II allowed us to study the cell cycle within a single root at cellular resolution. For example, in a 5 day old *A. thaliana* root, cortex precursor tissue between 0 and 200 µm (in which cells undergo mitosis) contained 258 cells, 49 of which had EdU-positive nuclei, and five were in mitosis, while the transition zone (200–400 µm from the QC, before rapid cell elongation) contained 217 cells with 48 cells with EdU-positive nuclei and no mitotic cells (Figure 4f,g). Similar data can also be extracted for all other layers. This kind of analysis has been used previously for quantifying root zonation in each cell type and led to the conclusion that RAM size is tissue-layer-specific [17,29].

Module II (in combination with Module I) analysis can also be used for stem-cell niche studies. The stem-cell niche is defined on the basis of QC cells, which are quiescent stem cells with very long G1 duration [14]. The long G1 duration of the QC cells might be dependent on their very dense nuclei (Figure 2e) that have a delay in DNA replication (Appendix A, see above). Hence, module I and II analysis in the stem-cell region using mutants and different environmental signals provides additional insight into the biology of plant stem-cell niche activity [30].

The rootward region of the root tip (including central columella and lateral RC cells) is not involved directly in root growth, but it plays an essential role in protecting the RAM against abrasive damage of the soil and in the perception of several environmental signals. This is a tissue with rapid turnover of short-lived cells that is regulated by an intricate balance of new cell production, differentiation, and programmed cell death [31]. The precise kinetics of cell production, DNA replication, and growth in the RC and columella has been studied with regard to WOX5 function [32]. Here, we show an example of the atlas of columella and RC for the cell-cycle events (Figure 4h,i).

Cortex nuclei in the mature zone of the alfalfa and tobacco PRs keep a regular structure (close to round nuclei with regular chromocenter distribution), which might account for pseudo-nodule induction in tobacco and nodule induction in Fabaceae (Appendix A). Detailed analysis of cell-cycle events in the differentiation zone of the root shows that DNA replication was also observed within the vascular cylinder. However, only pericycle cells can fulfill mitosis in this region during LR formation in *A. thaliana*.

Another unknown issue in root biology is how phytohormones regulate cell-cycle transitions, which is key to understanding plant growth, tissue regeneration, and stress responses [33]. The effect of the exogenous auxin on cell-cycle regulation is well known [34], but it has not been analyzed at a cellular resolution and whole-organ level. Analysis with module II will allow closing this gap. We found that short incubation with 100 nM of synthetic auxin NAA significantly inhibited DNA replication and cell division of all cells within the division zone of the RAM in *A. thaliana*, while it induced DNA replication of pericycle cells in the mature zone of the root, which is the source of new LRs (Figure 5). Interestingly, cells of the transition zone different from the pericycle and the vasculature did not replicate their DNA in response to NAA.

In conclusion, module II allows us to quantify cell-cycle events in situ for all cell types and calculate the kinetics of the cell cycle [13].

### 2.3. Module III: Immunolocalization, Protein Complex, and Protein/Protein Interactions

As inferred from the results of module I, each single cell within the RAM from the same or different layers displayed a specific nuclear morphology, which would affect their gene expression and protein activity levels. This, in turn, makes the classical molecular biology methods of protein-level quantification and analysis (i.e., Western blot) questionable and demands a robust method to determine protein localization and protein levels at single-cell resolution.

Protein localization in plant cells is commonly studied using fluorescent protein fusions [35]. However, this approach requires obtaining stably transformed plants, which is time-consuming and is limited to some plant species with available transformation and tissue culture protocols. In addition, the presence of a large hydrophilic region of the fluorescent protein may produce recombinant proteins with slightly different characteristics from the native proteins, which, in turn, can affect their spatial localization.

Protein immunolocalization with a custom antibody against the protein of interest is a good alternative to study protein localization, as the antibodies can be assayed directly in different genetic backgrounds, including mutants, without the need for further crossing. If the antibody design is adequate, cross-reactivity in multiple plant species is possible (e.g., PIN1 and tubulin antibodies [36,37] and histone variant antibodies). One additional advantage of protein immunolocalization is the possibility of performing triple or quadruple labeling and studying protein–protein interaction in situ. Commonly used methods of studying a protein complex by pulldown or co-immunoprecipitation require cell lysis that might change protein conformation; hence they may not exactly allocate subcellular protein complexes. Moreover, protein complex composition may also differ between neighboring cells. Hence, in situ detection of a protein complex may provide adequate information about its functional activity.

Module III includes a detailed protocol for double or triple labeling with different antibodies, as well as a proximity ligation assay (PLA assay) to study protein complex interactions [38]. As an example of module III output, Figure 6a,b show the immunolocalization of the auxin efflux facilitator membrane proteins PIN-FORMED1 (PIN1), PIN2, and PIN4 in the *A. thaliana* RAM. Quantitative analysis of the PIN1–PIN4 complex localization and levels in 3D has previously been presented and may explain the functional activity of the PIN1 and PIN4 protein under different treatments [39]. Another example of module III application is the localization of histone modification variants in the RAM, due to that fact that most of the histone antibodies that are commercially available recognize conserved peptides of both plant and animal histones. Figure 6c,d show an example of the H3K9me2 mark that has been suggested as a key hub for regulation of chromatin status and gene expression [40]. Nuclear segmentation analysis showed the colocalization of H3K9me2 with DNA replication and with the chromocenter. Other histone modifications can now be studied by module III.

### 2.4. Module IV: Cell Geometry

Size is a fundamental feature that governs the flow of all biological processes. However, cell size differs drastically across plants and cell types. In each of these cases, subcellular structures must be scaled according to the size of each cell and to their respective contribution to the cell function in plants.

That is why information on cell geometry is essential to understand the phenotype of plant organs. For example, root waving and gravitropic response phenomena are causally related to unequal cell elongation/expansion in the outer cell layers (cortex/epidermis) [41]. Therefore, a detailed characterization of the cells in each cell type is required to understand the waving mechanism of root growth.

For module IV analysis of *A. thaliana* and foxtail millet (*Setaria italica* (L.) P. Beauv.) PRs, individual cells were segmented and assigned to proper layers as described elsewhere [42] (Figure 7a,d). In both species, we found that cell volume increase was highly dependent on cell layer. The outer layers of the RAM (epidermis and cortex precursors) accounted for more than 80% of the total of the root volume (Figure 7b,e). Other cells, such as those of the stele region in the *A. thaliana* RAM, are characterized by their small contribution to total root volume and large numbers (Figure 7b). Moreover, the increase in the cell volume of neighboring cells along the longitudinal axis was region-specific, with higher values in the epidermis and cortex precursor cells (Figure 7c). Using module IV, detailed analysis of xylem/phloem volume/elongation for both *A. thaliana* and foxtail millet PRs could also be performed (Appendix A).

In the PR of C4 plants, such as foxtail millet, cell volume of the central metaxylem and protoxylem is extremely large, in opposing contrast to phloem cells (Appendix A).

Module IV also allows studying root asymmetry. In *A. thaliana* PRs, the epidermis precursors produce two different cell types: trichoblast (T), from which the root hairs are produced, and atrichoblasts (AT). Eight T cells connect with two cortex cells, while each AT cell connects with only one cortex cell. Generally, eight epidermis initials directly contact the QC cells in the stem-cell region and are the origin of all 20–24 epidermis precursors by tangential divisions occurring at variable spatial coordinates (Figure 8a,b). For a deeper analysis of cell features, we drew “unrolled root graphs” containing all cells of a given layer with regard to their radial location (*x*) and distance from the QC (*y*), which might also accommodate other parameters, such as cell or nuclear volume (Figure 8b). Unrolled root graphs from the same or different samples might be combined to create cellular density maps that allow a detailed study of cellular interactions at the tissue level (Figure 8c). Module IV analysis now allows the detailed mapping of these formative divisions in different genetic backgrounds and in response to different environments. The other rather unexpected feature of epidermal precursors in *A. thaliana* PRs is represented by the remarkable differences in cell length between presumptive AT and T cells (almost twofold longer for AT) but a similar cell volume. Several chromatin parameters on these cells could then be studied by applying module I analysis (Appendix A).

In addition, module IV allows measuring cell growth. This is very important, especially in *A. thaliana*, in which cell length does not reflect cell growth at all and leads to confusion in geometrical root zonation (Appendix A).

In addition, we applied module IV analysis to study the effect of exogenous auxin on the morphology of cells at the mature zone of the root, confirming the localized and temporal activation of some pericycle cells after a short incubation with 100 nM NAA (Figure 8d–f).

## 3. Discussion

Plant productivity relies on the correct functioning of different organs with specialized functions within the plant body, as well as on their interactions with biotic and abiotic factors. Plant organs consist of hundreds or thousands of cells from different tissues, with each cell having a given gene and protein profile and a fixed spatial allocation, which contributes with specific functions and cell-to-cell interactions. In addition, the shape and size of all plant organs are determined by the contribution of their individual cells through organized cell division and cell expansion patterns. Hence, a comprehensive description of individual cell behavior on a growing organ is crucial for our understanding of plant development. Recent advances in tissue labeling, microscopy, and bioimage analysis have led to the opportunity to build detailed 3D representations of plant organs, including morphological, genetic, and functional information in each cell.

Roots are key factors for anchoring the plant to substrate and for nutrient and water uptake, and their simple morphology and the development of tissue-specific fluorescent markers has allowed obtaining gene expression profiles of different root tissues in plant model species such as *A. thaliana* [43] and tomato [44]. This approach relies on the use of molecular markers, which are not directly available in non-model plants that normally require tissue culture techniques which are known to alter cellular interactions and might affect the expression of some genes. Lowering of next-generation sequencing (NGS) and bioinformatics costs has allowed the use of single-cell RNA-sequencing techniques [45] to characterize the spatiotemporal developmental trajectories of every cell within a single root [46,47,48].

Despite these extraordinary advances, many plant biology laboratories lack the appropriate expertise and/or funding to apply the state-of-the-art approaches for quantitative description of root features at cellular resolution. To close this gap, we described here a dedicated workflow to perform a 3D analysis of root features at single-cell resolution. This workflow has been designed along four phenotyping modules requiring minimal expertise and/or equipment.

Module I is aimed at nuclear and chromatin analysis and might be used to build chromatin accessibility maps when combined with ATAC-seq and laser-capture microdissection [49]. This module also allows studying cell differentiation status, as chromatin organization is tightly linked to gene expression regulation and to the differentiation status of the cell. We found that chromatin status in the proliferation zone of the RAM in *A. thaliana* varied between neighboring cells, even within the same layer (Figure 2).

Module II involves cell-cycle analysis aimed at a functional study of root zonation. Several authors defined the meristematic region of the PR in *A. thaliana* on the basis of relative elongation of cortex precursor cells, without considering other tissues or the actual proliferation activity of the cortex precursors [15,17]. Despite the practical value of this definition, application of modules II and IV has allowed us to precisely map cell division, cell endoreduplication, and cell elongation/growth in every cell [19], allowing the building of integrative 3D models of the RAM zonation in species with complex root architecture such as tobacco, alfalfa, or foxtail millet. Moreover, as shown previously, RAM length and cell-cycle kinetics are not constant throughout the root tissues [13,16], and each cell layer has its own cell-cycle kinetics, proliferation domain, and transition domain sizes.

Additionally, our pipeline will be useful to broaden the range of species to study LR initiation and nodule/pseudo-nodule induction, both occurring in the mature zone of the root. The information obtained by applying the proposed workflow to nonflowering plants, which allows underpinning key histological features during RAM evolution. Our results also indicated that, shortly upon auxin application, DNA replication occurred not only in the pericycle but in the neighboring procambium of the differentiation zone of the root, while endodermis and cortex cells of this region were unresponsive. By applying module I and II analyses to known auxin mutants and by applying different treatments (i.e., auxin biosynthesis/response inhibitors), the missing link between auxin function and cell-cycle regulation could be finally untangled.

Combination of modules I and II will allow investigating the relationship between cell-cycle stages and chromatin organization in each cell; hence, comparisons between different tissues and/or regions along the longitudinal root axis will be possible. Moreover, cell division occurs in cells of specific cell fate and certain chromatin organization, and detailed 3D investigation will allow us to understand the link between cell-cycle kinetics and cell fate/nuclear geometry. In addition, a comparison of PRs, LRs, and nodule formation in several species might reveal conserved features of proliferative plant tissues.

Module III includes a detailed investigation of protein localization, histone modifications, and protein complex formation. There are two main methods for the detection of protein localization: fluorescent recombinant protein and immunolocalization. The first one has several disadvantages, such as the requirement of gene cloning, protein fusion construction, and plant transformation. Moreover, introducing a fluorescent protein into mutant genotypes requires crossing, and it is highly time-consuming. The list of available antibodies with cross-reactivity in other plant species has increased considerably [50]. As extra benefits, the possibility of colocalization, triple labeling, and protein complex investigation [38] makes this module very attractive for researchers interested in protein complex studies.

Lastly, module IV analyzes cell geometry features. The size and shape of a given plant organ is ultimately determined by the contribution of its constituent cells through precise division and expansion patterns on a 3D space. As cell size differs drastically across plant organs and cell types, growth coordination at the tissue and organ levels is required [51]. To understand the phenotype of a complex plant organ, such as the root, information about the geometry and spatial coordinates of all its cells needs to be obtained. For example, root waving and gravitropic response are causally related to unequal cell elongation in the outer cell layers of the epidermis and cortex [41]. By applying module IV to the study of roots of several plant species (*A. thaliana*, tobacco, alfalfa, and foxtail millet), we will be able to determine the conserved features of the RAM structure, such as the stem-cell niche region and the extent of formative regions in the division zone of the RAM.

## 4. Materials and Methods

### 4.1. Plant Material

Seeds of *Arabidopsis thaliana* (L.), Heynh, *Medicago sativa* L., *Nicotiana tabacum* L., *Setaria italica* (L.) P. Beauv., and the growth conditions used were thoroughly described elsewhere [19,29,37,52].

### 4.2. Plant Material Preparation and Labelling

All procedures for plant cultivation, fixation, treatments, and imaging were described previously [19,29,37,52]. The main differences from the previously published protocols are presented below.

#### 4.2.1. Cell Boundary Labeling

Our cell boundary labeling protocol is based on the binding of PI to deketonized cell-wall polysaccharides at low pH (1.4) in the presence of sulfur. Although the basic protocol was described previously [36], significant modifications were required to adapt the protocol for 3D scanning and analysis of the inner tissue layers, including xylem and phloem tissues. One limitation is that the original fixation in acetic acid leads to significant tissue maceration and often damages the softer mature plant parts. For this reason, we recommend fixation with formaldehyde in microtubule-stabilizing buffer (MTSB) at pH 7.0 before labeling. The deketonization level (time of periodic acid treatment) is another crucial parameter. For *A. thaliana* roots, a 30 min deketonization in 1% periodic acid partially punctured the cell walls, especially in the vasculature with thinner cell walls. We recommend reducing the deketonization time to 15–20 min. The mounting procedure is another crucial step. A thick object can be scanned using double-sided scanning by mounting the samples between two coverslips used as a spacer: 24 × 60 mm size as a base and a 24 × 32 mm size as a cover. The spacer should have a similar thickness to the object. This adjustment allows the object to be scanned from both sides to avoid a low signal-to-noise ratio in the deeper parts.

#### 4.2.2. Chromatin and Cell-Cycle Detection

For detection of the chromatin and cell cycle, plants were transferred to a 12-well plate with appropriate liquid medium (TK1 for *A. thaliana* and TK4 for the other species used [53]) for 12 h. Thereafter, 10 µM EdU was added for appropriate incubation period. Seedlings were then fixed and subjected to detection EdU detection and scanning as described [36]. For nuclear structure, we highly recommend scanning images with voxel size 0.2 × 0.2 × 0.3 (*XYZ* resolution). Dynamical corrections of the scanning also require keeping the signal intensity and signal-to-noise ratio as constant as possible through whole stacks. To achieve this aim, one may adjust the detector gain and the average scan numbers for each sample/section.

#### 4.2.3. Preservation of the 3D Structure

Preservation of the 3D structure is crucial for properly analyzing the cell and nuclear shape and cell position. The most sensitive issue is shape alteration in mature cells with large vacuoles that can quickly shrink under non-physiological conditions. We used several tricks to prevent shape disturbing. We minimized the time of deketonization and cell-wall digestions to keep the cell wall thinner. However, the most crucial steps were the methanol treatment and the mounting procedure. For the methanol, we suggest adding water to methanol very gradually rather than transferring samples to a new solution. Direct transfer to mounting medium after labeling eventually led to a disturbance of cell shape. To avoid this issue, we gradually added 50% glycerol to the samples before mounting them to the final concentration (25% glycerol) in the mounting medium. This trick allows keeping the original shape of the cells even in the differentiation zone of the root.

#### 4.2.4. Image Analysis

Image analyses were performed essentially as described earlier [42,54]. Briefly, images were converted to hdf5 format using the LOCI plugin for ImageJ, and then stitched together to obtain a root tip total length of 400 μm from the QC using xuvTools [55]. Finally, 5–10 representative roots were chosen for detailed annotation. The DAPI and EdU channel images were processed with the iRoCS Toolbox [25] in the following way: nuclei were automatically detected using the “01-Detect Nuclei” plugin, and then the epidermis was semiautomatically labeled using the “02- Label Epidermis” plugin. After the QC position was marked (Channel -> New Annotation Channel), the nuclei were set in spherical coordinates using the “03-Attach iRoCS” plugin. Automatic classification of the nuclei to the corresponding cell types (epidermis, endodermis, cortex, pericycle, vasculature, and root cap) was done using the “04-Assign Layers” plugin, which also enabled the automatic annotation of nuclei in mitotic state (option “reclassify mitotic state”). All annotated roots were manually corrected for erroneous layer, mitosis, or EdU assignments.

#### 4.2.5. Nuclear Segmentation

For the nuclear geometry analysis, original nucleus-labeled images were converted to .tiff format, and OTZU methods were applied [56]. Thereafter, nuclear edges were detected, images were converted to hdf5 format, and further segmentation was performed accordingly to the standard pipeline used for cell segmentation [42].

#### 4.2.6. Individual Nuclear Analysis

Individual nuclear analysis was performed accordingly to Dubos et al. [26] with small modifications. First, we selected 30–40 sections that contained a similar intensity of nuclear labeling with fewer than 100 nuclei. Second, auto crop was performed with 6 × 6 × 7 µm settings (*XYZ* resolution). The subsequent steps were performed as described previously [26].

## 5. Conclusions

The workflow proposed for in situ quantitative root biology will allow obtaining detailed 3D maps of nuclear and chromatin organization in the RAM of non-model plant species without the need for molecular markers. Using different DNA fluorescent dyes, cell-cycle stages could also be measured at single-cell resolution. By combining several of the phenotyping modules described here, it will be possible to characterize the different regions of the RAM (i.e., root zonation) with unprecedented resolution.

Our four-module phenotyping pipeline will allow researchers working in the root biology field to enlarge their cell biology toolset, and we do not exclude that other plant researchers working in shoot apical meristem and leaf initiation or de novo organ regeneration will apply our method in their investigations.

## Figures and Tables

**Figure 1 plants-10-02399-f001:**
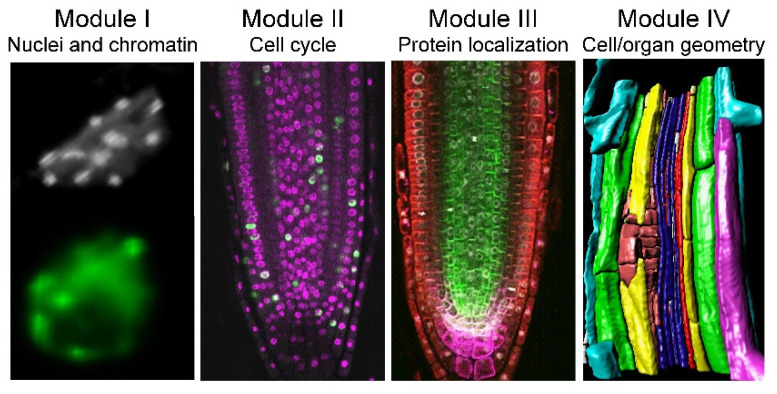
Proposed workflow for in situ quantitative root biology. We defined four different modules for cellular phenotyping of serial confocal images. Each cell within a single root is defined by a 3D coordinate system [25], and the proposed method allows the simultaneous extraction of cell geometry, cell cycle, and chromatin features.

**Figure 2 plants-10-02399-f002:**
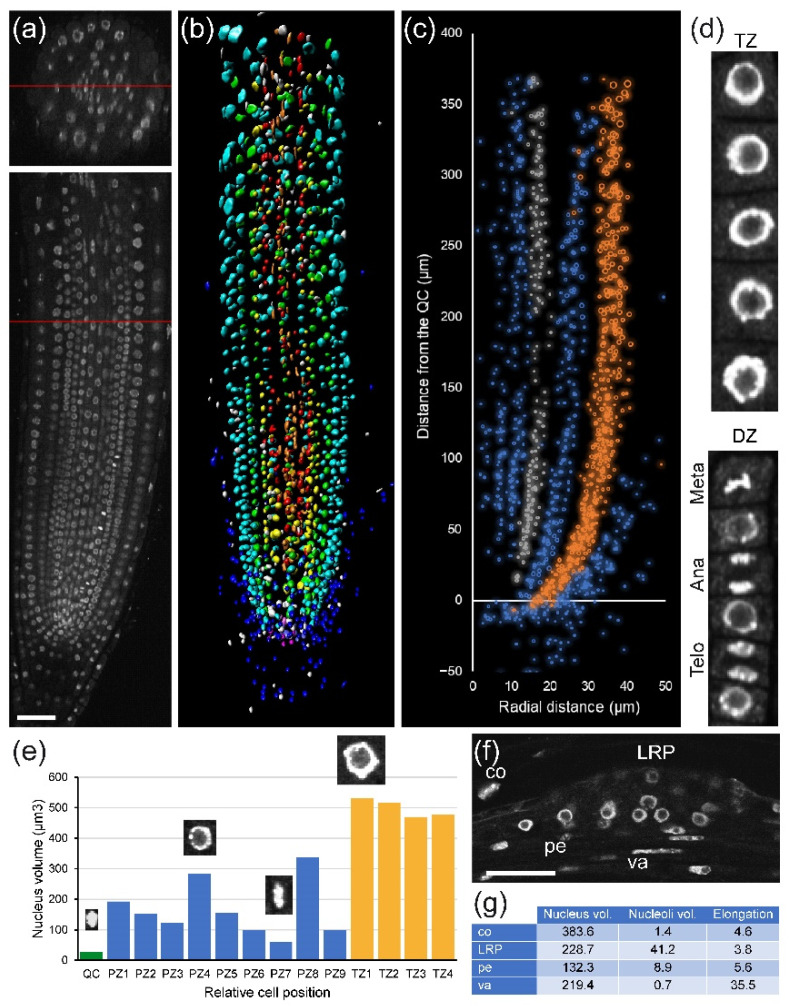
Analysis of the chromatin structure after propidium iodide (PI) labeling in *A. thaliana* PRs. (**a**) Confocal 3D image of a PR; top: cross-section, bottom: longitudinal section; red lines indicate where the sections were taken. (**b**) Overview of the nuclear volumetric analysis; each color represents nuclei from same layer. (**c**) Nuclear distribution with radius as cell volume along the longitudinal and radial axes of different layers; epidermis precursor cells are depicted in orange, and pericycle initials are shown in gray. (**d**) Selected nuclei of epidermal precursor cells in the division zone (DZ) and the transition zone (TZ) of the RAM for detailed analysis. Meta: metaphase, Ana: anaphase, Telo: telophase. (**e**) Nuclear volume of individual cells. QC: quiescent center. Some images of nuclei are shown. (**f**) A representative 2D image after DAPI labeling in a newly formed LR primordium (LRP). Individual cells used for data extraction are shown (co: cortex, pe: pericycle, va: vascular). (**g**) Quantitative data from individual cells. Volumes are in µm^3^. Scale bars: 50 µm.

**Figure 3 plants-10-02399-f003:**
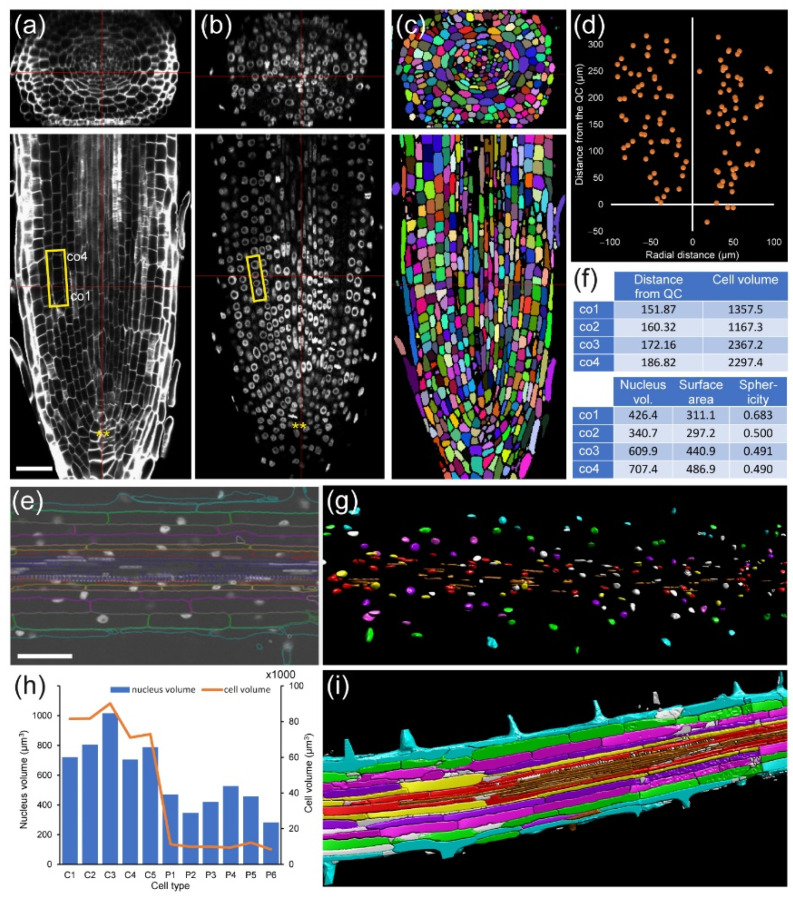
Nuclear and chromatin analysis in *M. sativa* and tobacco roots. Roots were fixed, double-labeled for cell boundaries and nucleus, scanned, and analyzed as described in Section 4. (**a**) Cell wall and (**b**) nuclear labeling in the PR of *M. sativa*. Yellow rectangles in a, b: cells and nuclei chosen for detailed analysis. (**c**) Cell labelling and cell segmentation. (**d**) Mitosis distribution along the longitudinal and radial planes in all root tissues. (**e**) Details of morphological features of selected cells and nuclei from a, b. Distances are in µm, areas are in µm^2^, and volumes are in µm^3^. (**f**) A representative 2D image in the mature region of a tobacco PR. (**g**) Nuclear segmentation of the PR shown in f; different colors indicate nuclei from the same cell layers. (**h**) Nuclear volume and cell volume representation in individual cells in the division zone of the RAM; C: cortex precursor, P: pericycle initial. (**i**) 3D rendering of the cell boundary labeling and segmentation. Each color represents individual cell layers. Scale bars: 50 µm.

**Figure 4 plants-10-02399-f004:**
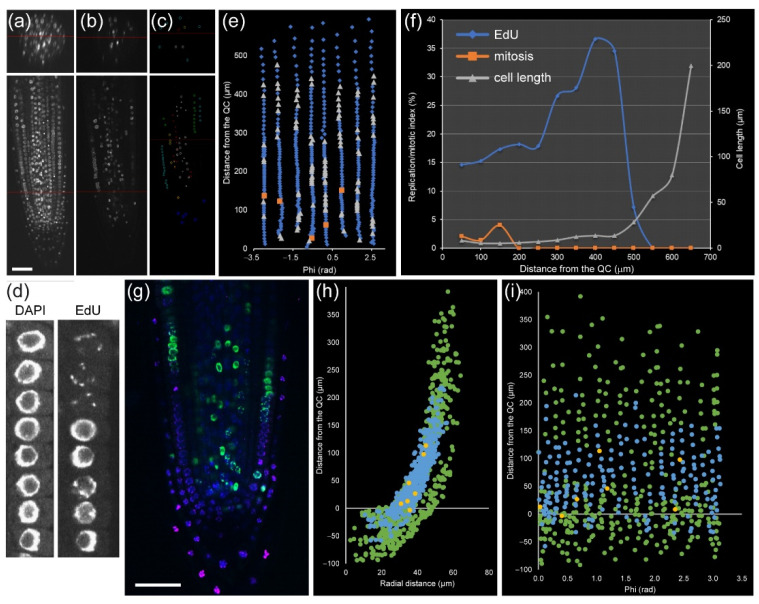
Detection of key cell-cycle events in *A. thaliana* PR. (**a**) DAPI labeling for nuclear and mitosis identification. (**b**) EdU staining (20 min) for detection of DNA replicative cells. (**c**) EdU-positive nuclei segmented by cell layer. (**d**) DAPI and EdU staining on cortex precursor cells in the RAM. (**e**) Unrolled layer of cortex precursor tissue. Blue: interphase nuclei, gray: DNA replicative nuclei, orange: mitotic nuclei. (**f**) Quantitative analysis of DNA replication (blue) and mitotic index (orange) in the cortex precursor tissue along the longitudinal axis. Cell lengths are in gray. (**g**–**i**) DNA replication in the root tip. Seedlings were incubated with EdU for 90 min (**g**) (DAPI: blue, EdU: green, PI: pink) and for 8 h (**h**) (DAPI: green, EdU: blue, mitotic nuclei: yellow). (**i**) Unrolled layers of RC. Green: all nuclei; blue: EdU-positive nuclei; yellow: mitotic nuclei. Scale bars: 50 µm.

**Figure 5 plants-10-02399-f005:**
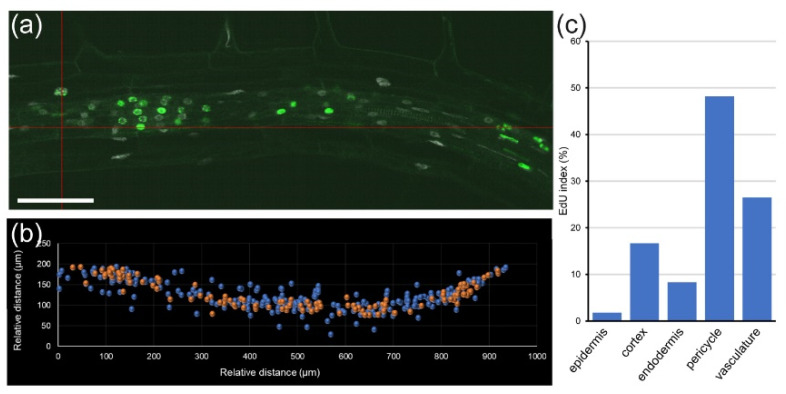
Effect of auxin on cell-cycle events in *A. thaliana* PRs. Seedlings were treated for 16 h with 100 nM NAA, EdU was added for 90 min, and samples were fixed and analyzed as described in Section 4. (**a**) Confocal 2D image; EdU-labeled replicating nuclei are green, DAPI-labeled nonreplicating nuclei are white. (**b**) Position of the DAPI-labeled nonreplicating nuclei (blue) and EdU-labeled replicating nuclei (orange). (**c**) Cumulative EdU index in different cell types. Scale bar: 50 µm.

**Figure 6 plants-10-02399-f006:**
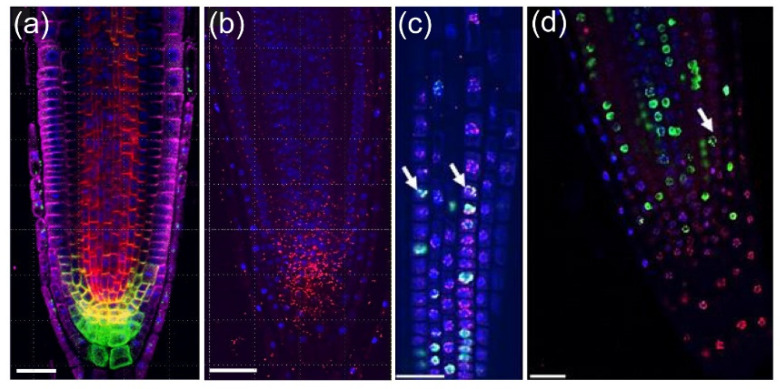
Protein and protein complex localization in *A. thaliana* roots. (**a**,**b**) Detection of PIN1 and PIN4 complex by proximity ligation assay in *A. thaliana* PR. (**a**) Immunolocalization: PIN1 (red), PIN2 (magenta), and PIN4 (green). (**b**) Proximity ligation assay: PIN1–PIN4 complex (red). Nuclei are stained blue with DAPI. (**c**,**d**) Colocalization of the H3K9me2 with DNA replication events. DAPI staining is shown in blue, EdU (90 min incubation) staining is shown in green, and H3K9me2 immunolocalization is shown in magenta. Arrows mark the visible colocalization of DNA replications with H3K9me2. Scale bars: 20 µm.

**Figure 7 plants-10-02399-f007:**
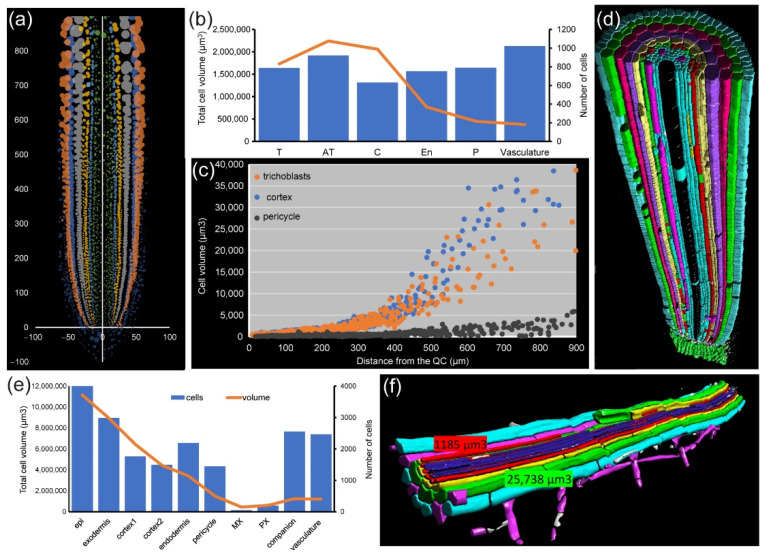
Cell geometry features in *A. thaliana* and foxtail millet PRs. (**a**) Unrolled *A. thaliana* root with distance from the main axes. (**b**) Cell number and cell volume in the first 1 mm from the QC. Bars: cell number; lines: cumulative volume. (**c**) Cell volume distribution along the axis in the trichoblast, cortex precursors, and pericycle initials. (**d**) A 3D render of foxtail millet root after segmentation. Cells are color-coded according to tissue layer. Unsegmented cells are depicted in black. (**e**) Cell number and cell volume in first 1 mm from the QC. Bars: cell number; lines: cumulative volume. (**f**) A 3D rendering of the mature part of the *A. thaliana* PR after segmentation. Numbers denote the cell volume in the cortex precursors and pericycle initials.

**Figure 8 plants-10-02399-f008:**
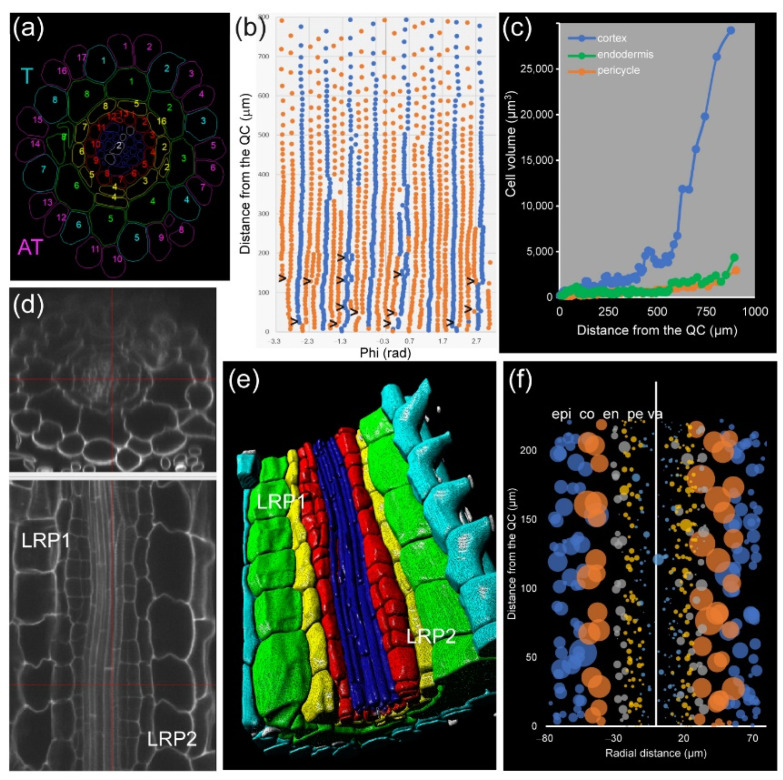
Study of cellular interactions at the tissue level in *A. thaliana* PRs. (**a**) Virtual cross-section of the PR at the transition zone with the cell numbers in each cell type; T: trichoblasts, AT: atrichoblasts. (**b**) 2D graph on an “unrolled epidermis” (blue: T; orange: AT); tangential cell divergence shown by >. (**c**) Evaluation of the cell volume in single cells of cortex precursors, endodermis precursors, and pericycle initials. (**d**–**f**) Auxin-induced (100 nM NAA during 18 h) LR primordia (LRP). (**d**) Original 3D scan indicating the position of two LRP (stage II). (**e**) 3D rendering after segmentation. (**f**) Radially unrolled root with cell volume as radius of the circle; epi: epidermis, co: cortex, en: endodermis, pe: pericycle, va: vasculature.

## Data Availability

The data presented in this study are available on request from the corresponding author.

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
