# Peer review of "Methods of In Situ Quantitative Root Biology"

_plants, 2021, doi:10.3390/plants10112399_

Round 1

Reviewer 1 Report

In the presented manuscript titled “Methods of in situ quantitative root biology”, the authors summarize and review previously developed methods for root research without using transgenics. All methods described are nowadays considered to be standard methods in Arabidopsis thaliana model organism as the freshest method was described more than ten years ago. Moreover, the methods have been also used in other plant species.

Except for a minor suggestion in optimizing the existing methods, the manuscript is lacking novelty. The most exciting part of the manuscript is the image analysis approach. I well appreciate the imaging effort and beautiful models’ representation. However, the image analysis approach has been explained elsewhere as the authors cited. I reckon the manuscript is not an original research article, but rather a review article.

With a slight format change into a review article, I agree that the manuscript might serve as a reference “…for researchers who need to improve their quantitative root biology portfolio...”.

Major concerns:

The manuscript would benefit from adding a definition of a meristem. The meristem is composed of mitotically dividing cells that are gradually transitioning from a stem cell state into a differentiated one. Once a cell stops dividing, it leaves the meristem and finalizes its differentiating program [Wendrich et al., PNAS 2017 “Transcriptional gradients in Arabidopsis roots”]. This would improve the reading of sentences in lines 29-30, line 60, and lines 213-214 (see minor corrections for more details).

Line 60: “…In addition to its protective and signaling role, the distal meristem…” Meristem cannot have a protective role, but cells produced from the distal meristem can. Please revise this sentence.

Although the authors pose very relevant questions in the introduction part, they do not give answers to them in the results section. For example, they raise the problem of meristem size quantification (in lines 55-57) and say that they provide methods to resolve the problem, however, they do not offer the results that prove or discard their claims and thus the method relevance.

Alternatively, the result statement is on occasion very strong without real data justification. For example, in lines 125-127, the authors claim to distinguish different cell cycle stages, while in Figure 2D-E one can only distinguish interphase from cells in mitosis with certainty. Similar in the lines 149-151, “…we clearly distinguished different cell cycle stages (G1, G2 and mitotic cells) ...” but in the figure (3A-E) and its legend, one can only distinguish different stages of mitosis from the interphase.

The results are sometimes only offering the uninformative image while the conclusion is drawn from a cited literature. For example in the line 151: “Next, we applied module I analysis to tobacco (Nicotiana tabacum L.) roots.” The image is shown in Figure 3F or 3G. However, the conclusion is made from a citation: “In contrast to that found in Arabidopsis, cortex cells in the mature region of the tobacco PR can divide and even induced the pseudo-nodules [25].” Such a result is not visible in Figure 3F-G. Thus, this result is uninformative.

Line 208-210: “The long G1 duration of the QC might be dependent on their very dense nuclei (Figure 2E) …” Please comment here on the method limitations due to resolution obtained from conventional wide-field and confocal microscopes when it comes to the cells with smaller volume.

Line 222: Xylem cells in differentiating part of the Arabidopsis root undergo programmed cell death. Thus, it is very unlikely that DNA replication would occur once a secondary cell wall is formed. In Figure S3 the green (EdU labeled cells) may instead mark procambial cells, which in the mature part of the root may proliferate. Later in the development, this proliferation is a basis for so-called secondary organ growth. While the image is not clear, please rephrase “…including xylem cells replicate their DNA” into “DNA replication was observed within vascular cylinder”.

Line 230/231 The effect of the exogenous auxin is well known. Examples are in the review of Perrot-Rechenmann (2010) Cold Spring Harb Perspect Biol. Please amend the statement.  

Quantitative results were mostly represented without standard error/deviation and devoided of any statistical analysis. Please, include it. I understand that in lines 199-203 (Figure 4G) the authors are only giving an example, but the result would be more appreciated if the method/pipeline can easily process multiple roots and give an average representative situation of a biological system.

Minor corrections:

Line 27 & 28: Both sentences start with “The Root”. Please revise the article usage.

Line 29: “… hereafter Arabidopsis), it is composed…” I suggest there could be two sentences, the second starting with underlined sentence part.

Line 29/30: “…distal meristem (columella and root cap) …” Columella and root cap are differentiated cell types; thus, they do not qualify as a meristem. The distal stem cells produce columella and lateral root cap cells. I suggest a correction “…distal meristem (producing columella and root cap cells)”

The manuscript is mostly well written, although it may benefit from certain paragraph rearrangement in order to improve the flow. As an example, the paragraph stretching from line 63 to 69 flows better after line 48. Sentences line 58-62 appear without previous connection and disturb the reading flow.

Lines 151-152: “Next, we applied module I analysis to tobacco (Nicotiana tabacum L.) roots. In contrast to that found in Arabidopsis…” What is “that” referring to?

Lines 154-157 “Detailed nuclei structure analysis shown that cortex cells in tobacco contained round nuclei with similar sphericity (while in Arabidopsis cortex cell have a very irregular chaotic structure with small nucleoli). These features were like those to the pericycle cells (Figure 3F-I).” This figure does not show results for Arabidopsis, thus please add a figure calling (I assume it is in supplemental data) or a reference.

Line 213/214: Similar to lines 29/30 and line 60, meristem cannot serve a protective role, these cells are dividing and fragile. Please rephrase. Also here, for the third time, there is a misleading repetition of what is distal meristem. Please reduce redundancy and correct the sentence.

Line 235/236: “Interestingly, these cells of the transition zone…”  Here, it is not clear what cells are in the matter. Please clarify.

Letters labeling panels in figures should match figure legends and figure calling in the manuscript (capital or small) letters.

Figure 4: please correct the panel calling and figure legend.

Reviewer 2 Report

The paper is a hybrid between a research paper and review. As such the methodology is not detailled enough to understand how the results were obtained and the reader is frequently refered to papers by the same researchers. I think the paper whould benefit if more detail is supplied in the methods, especially considering that this paper is supposed to be a technical note. Would someone not skilled in the techniques be able to repeat the work based on the details provided?

I am wondering if any independent validation has been performed on the results? The authors present impressive images, but are the data shown correct? For instance in L390-392, the authors state "We found that chromatin status in the proliferation zone of the RAM in Arabidopsis roots varied between neighboring cells, even within the same tissue layer." Is this due to noise, or are these meaningful difference? If so, what would cause the differences?

The data procressing steps need to be explained in more detail- how have different layers been stitched together? How have the 3D images been constructed?

Apart from the points, there are a few instances when the writing is not clear, e.g. L350-352, 390-392, L430-431

Overall an impressive paper, which was well written and relatively easy to read.

Reviewer 3 Report

Dear Autors,

The manuscript "Methods of in situ quantitative root biology" submitted for review is very interesting and useful, therefore I expect it to be published - after corrections, because it cannot be accepted in its current form.
The manuscript has two main deficiencies:
1. You claim to provide new protocols - verbatim in the abstract: "Detailed protocols are also provided."; if so, the said protocols must be specified step by step to be "detailed". However, they are missing, instead, in the M&m chapter there are references to three earlier papers (reporting the studies performed in different laboratories by different people and at different times, so the respective original protocols are not completely consistent) plus information about modifications done by you. It is impossible to understand, what you actually did without opening several documents and collating them, and next, collating them with the present M&M chapter. If you do not want to expand the manuscript, you should - you must ;-) - put the step-by step protocols in a supplementary file.
2. You write in a manner, which in multiple places make the text illogical/ imprecise/ confusing due to the use of inappropriate terminology or mental shortcuts (besides: such imprecisions are very confusing especially for students, obviously inexperienced - and I claim this as an old academic teacher). The scientific language must be precise and unequivocal, even if it results in it being somewhat dry and boring ;-) A specific term must refer to a single "thing" only, and you apply freely to the root apical meristem the terms appropriate to the older parts of the root. I am fully aware that these and similar errors are commonly committed, but their universality absolutely cannot be an excuse for leaving them in the manuscript under review. The text must be carefully scrutinized and any imprecision must disappear (please).

All my comments are marked in the text of the manuscript, which I attach to the review and which is an integral part of it.

I am a bit sorry that in this letter I summed up the positive aspects of the manuscript with three words in the first sentence, so at the end I would like to emphasize strongly that I like the manuscript and it is needed.
With best regards,
Sincerely yours,
Reviewer

Author Response

We are tremendously thankful to the reviewer for its extensive editing and suggestions. Most (if not all) of them were highly relevant, and we tried to implement them in the new version of the ms. after extensive re-writing. We provide individual answers to all comments in the attached PDF. This reviewer could check whether we appropriately answered all its request in the Word documents with the tracked changes on.

There is only one request that we have not addressed, and it is the one related to the nomenclature used to refer to dividing cells of the RAM that are already committed (because of its position) but are not fully differentiated. At this point, we sticked to the nomenclature used in the Arabidopsis thaliana field, which has been established earlier (Dolan et al. 1993) and it is widely accepted (Petricka et al. 2012). According to this nomenclature, we refer to tissue initials or stem cell initials and dividing cells within the RAM as, for example, pericycle initials or pericycle stem cells and pericycle cells of the RAM. Pericycle cells outside the RAM will then be referred as pericycle cells of the mature zone of the root.

Round 2

Reviewer 1 Report

I believe that the current version of the manuscript is much improved regarding scientific soundness and flow. I would like to suggest this review for acceptance to be published in Plants.

Author Response

Thank you very much for all your suggestions and comments that havel helped us a lot to polish an earlier version of this ms.

Reviewer 3 Report

Dear Authors,

please find below the copy of my review letter to the Editor.

With best regards,
Sincerely yours,
reviewer

The manuscript "Methods of in situ quantitative root biology" was reviewed by me in the 1st round with my recommendation of "major", next it was corrected by the Authors and submitted for the 2nd review round. As previously, I think the manuscript is very interesting and useful, but it cannot be accepted in its current form.
The Authors did much to improve the manuscript, and I appreciate their efforts. However, they did not respond satisfactorily to my two main concerns.  

In my opinion, the manuscript had - and still has - two main deficiencies:
1. 
The manuscript does not report on any new scientific discoveries. However, it is a Technical note, and its supposed value lies not in the new discoveries, but rather in the proposal of a comprehensive workflow for the quantitative microscopic analysis of the root tip, involving a few well-known visualization techniques and successive image processing and image analysis. The problem is, the Authors do not provide specific comprehensive protocol(-s). Instead, in the M&m chapter there are:
a) references to several earlier papers reporting the studies performed in mostly different laboratories by different people (not the Authors of the present manuscript, mostly, and at different times, so the respective original protocols are not completely consistent),
b) information about modifications done by Authors. 
Therefore, it is impossible to understand what the Authors actually did without opening several documents and collating them, and next, collating them with the M&M chapter of the present manuscript. Even after such work, the results would be very frustrating: the analysis of the references chosen by the Authors as basic for the methodology (the numbering below is as in the manuscript) shows that in most cases, the references do not include a ready-to-use protocol(s), and in some cases the reference does not contain the claimed method at all. 

* for plant material, cultivation, fixation, treatments and imaging:
19. Lavrekha VV, Pasternak T, Ivanov VB, Palme K, Mironova VV. 3D analysis of mitosis distribution highlights the longitudinal zonation and diarch symmetry in proliferation activity of the Arabidopsis thaliana root meristem. The Plant Journal 2017, 92, 834-594.
A. thaliana is the object only, the chapter Experimental procedures (equivalent to M&m) is very brief and thus lacks details, instead multiple references to earlier papers are cited. This is not acceptable, as the reader has to track details in a chain of inconsistent papers by different authors. 

29. Pasternak T, Haser T, Falk T, Ronneberger O, Palme K, Otten LA. 3D digital atlas of the Nicotiana tabacum root tip and its use to investigate changes in the root apical meristem induced by the Agrobacterium 6b oncogene. The Plant Journal 2017, 92, 31-42.
Exactly as above, concerning N. tabacum

37. Omelyanchuk NA, Kovrizhnykh VV, Oshchepkova EA, Pasternak T, Palme K, Mironova VV. A detailed expression map of the PIN1 auxin transporter in Arabidopsis thaliana root. BMC Plant Biology 2016, 16, 5.
Exactly as for the ref.19, only the descriptions are even briefer.

57. Otsu N. A threshold selection method from gray-level histograms. IEEE Transactions on Systems, Man, and Cybernetics 1979, 9, 669 62-66.
I just wish all the best to any average biologist who would like to understand (and apply) the math that fills the whole Formulation (equivalent to M&m) chapter... Besides the paper does not suit the subchapter, as it concerns the thresholding, which belongs to the image processing rather, which is the next step after the imaging.

* for cell boundary labeling
36. Pasternak T, Tietz O, Rapp K, Begheldo M, Nitschke R, Ruperti B, Palme K. Protocol: an improved and universal procedure for whole mount immunolocalization in plants. Plant Methods 2015, 11, 50.
Excellent immunolocalization protocol for A. thaliana, briefly checked also for Medicago sativa, Triticum aestivum, Lycopersium esculentum and Hedera helix. However, the visualization of the cell walls is a small fragment of the protocol and it would be not a fast task ;-) to separate the necessary equipment and reagents and "chores" from the whole. 

* for chromatin and cell cycle detection (incl. EdU test)
36. Pasternak T, Tietz O, Rapp K, Begheldo M, Nitschke R, Ruperti B, Palme K. Protocol: an improved and universal procedure for whole mount immunolocalization in plants. Plant Methods 2015, 11, 50.
As above. However, there is nothing on EdU test. Moreover, the protocol stops at "Step 9: Mounting", it does not include confocal scanning or image processing, or analysis.

54. Pasternak TP, Ruperti B, Palme K. A simple high efficiency and low-cost in vitro growth system for phenotypic characterization and seed propagation of Arabidopsis thaliana. bioRxiv 2020, doi: 10.1101/2020.08.23.263491.
In this paper, an excellent protocol is provided, but only for in vitro growing of A. thaliana, from seed germination and seedlings till (high scale) seed production. Nothing more, certainly not on the detection of chromatin and cell cycle, esp. using the EdU test.
Moreover, this is a bioRxiv paper, and bioRxiv is an online archive and distribution service for unpublished preprints in the life sciences. Articles are not peer-reviewed, edited, or typeset before being posted online. Once posted on bioRxiv, articles are citable and therefore cannot be removed. As indicated, "This article is a preprint and has not been certified by peer review." and "Readers should therefore be aware that articles on bioRxiv have not been finalized by authors, might contain errors, and report information that has not yet been accepted or endorsed in any way by the scientific or medical community."

* for image analysis
42. Pasternak T, Falk T, Paponov I. Deep-resolution plant phenotyping platform description. protocols.io 2021, doi: 10.17504/protocols.io.brsdm6a6
This reference is very good, no complaints ;-) 

53. Tang S, Shahriari M, Xiang J, Pasternak T, Igolkina AA, Aminizade S, et al. The role of AUX1 during lateral root development in the domestication of the model C4 grass Setaria italica. bioRxiv 2021, doi: 10.1101/2021.05.20.444970.
The preprint contains a brief only protocol for S. italica growth conditions, sampling, fixation, and preparation for visualization of its root tip structure using propidium iodide + image processing and analysis. However, the analysis description is even more insufficient as it is summarized in just a single sentence ("Thereafter roots were segmented, layers were manually corrected, iRoCs were attached, masks were converted to markers and cell features were extracted to csv/xls files.").
Moreover, it is another bioRxiv preprint - my concerns as above.

* for nucleus segmentation - geometry analysis
42. Pasternak T, Falk T, Paponov I. Deep-resolution plant phenotyping platform description. protocols.io 2021, doi: 10.17504/proto-638 cols.io.brsdm6a6
as above conc. the same ref.

57. Otsu N. A threshold selection method from gray-level histograms. IEEE Transactions on Systems, Man, and Cybernetics 1979, 9, 669 62-66.
as above conc. the same ref.

* for individual nuclei analysis
26. Dubos T, Poulet A, Gonthier-Gueret C, Mougeot G, Vanrobays E, Li Y, et al. Automated 3D bio-imaging analysis of nuclear organization by NucleusJ 2.0. Nucleus 2020, 11, 315-329.
This reference is very good, no complaints ;-) 

My summary of the above: my previous opinion stands - the Authors are specifically asked to provide a ready-to-use comprehensive protocol for the whole workflow they propose. I am sure it would significantly increase the "citability" of the paper. If the Authors do not want to expand the manuscript, they can put the step-by step protocol(s) in a supplementary file.

2.
From my previous review: "The Authors write in a manner, which in multiple places make the text illogical/ imprecise/ confusing due to the use of inappropriate terminology or mental shortcuts (besides: such imprecisions are very confusing especially for students, obviously inexperienced - and I claim this as an old academic teacher). The scientific language must be precise and unequivocal, even if it results in it being somewhat dry and boring ;-) A specific term must refer to a single "thing" only, and the Authors apply freely to the root apical meristem the terms appropriate to the older parts of the root (additionally, they apparently do not discriminate between tissues and anatomical zones - root cortex or root stele are anatomical zones composed of a few tissues each). I am fully aware that these and similar errors are commonly committed, but their universality absolutely cannot be an excuse for leaving them in the manuscript under review. The text must be carefully scrutinized by the Authors and any imprecision must disappear (please)."
and from my comment in the 1st version of the manuscript:
"There is no cortex or epidermis (rhizodermis!) or endodermis etc within the meristem: the cells and tissues differentiate at best, and the mentioned tissues are not present yet! Stick to the meristem terminology rather than mental shortcuts ;-) , do not use the names of mature tissues to name the meristem domains. If you are not familiar with the histogens functional in the particular species (their set depends on the RAM organization), the protodermis, pre-cortex, proendodermis, procambium are always safe terms."
The Authors replied as follows:
"We agree with the reviewer that cells within the RAM are not fully differentiated, although there is a general consensuous (at least in the A. thaliana field; https://doi.org/10.1146/annurev-arplant-042811-105501) to use the same term when referring to all cells of the same tissue (with the exception of the initials or the stem cells (i.e. pericycle cells and pericycle initials or pericycle stem cells)." 
Thus the Authors replied with an explanation, but explanation is not equal to an justification.
My earlier opinion stands: the common error is still an error and it has to be corrected. The Authors are specifically asked to use the classical RAM terminology. 

Round 3

Reviewer 3 Report

Dear Authors,

I appreciate and accept your corrections of the manuscript, although I am (obviously) not happy with your position on RAM terminology issues. With all due respect to Dolan's (and others) research, his neglect of established terminology should not be maintained, especially since A. thaliana does not exhaust all the anatomical diversity and complexity of plants. 

All the best :-)

Sincerely yours

Rewiewer

Author Response

Dear Reviewer:

We would like to acknowledge again your extensive editing and suggestions, which have helped us to substantially improve our previous versions of this manuscript. We have now revised the text according to the established terminology in the field and we included the terms “precursor”, “initials” or “presumptive” to refer to cell in the RAM that are committed to a specific tissue but that are not fully differentiated yet. E.g., cortex precursor cells (instead of cortex cells), pericycle initials (instead of pericycle), presumptive trichoblast (instead of trichoblast), etc.

Sincerely Yours,

José Manuel Pérez-Pérez
Professor of Genetics
